# BRENDA-Score, a Highly Significant, Internally and Externally Validated Prognostic Marker for Metastatic Recurrence: Analysis of 10,449 Primary Breast Cancer Patients

**DOI:** 10.3390/cancers13133121

**Published:** 2021-06-22

**Authors:** Manfred Wischnewsky, Lukas Schwentner, Joachim Diessner, Amelie de Gregorio, Ralf Joukhadar, Dayan Davut, Jessica Salmen, Inga Bekes, Matthias Kiesel, Max Müller-Reiter, Maria Blettner, Regine Wolters, Wolfgang Janni, Rolf Kreienberg, Achim Wöckel, Florian Ebner

**Affiliations:** 1FB Mathematik u. Informatik, Universität Bremen, Bibliothekar. 1, 28359 Bremen, Germany; wischnewsky@escience.uni-bremen.de (M.W.); regine_wolters@hotmail.com (R.W.); 2Frauenklinik Universität Ulm, Prittwitzstr. 43, 89081 Ulm, Germany; office@gynova.at (L.S.); Amelie.deGregorio@uniklinik-ulm.de (A.d.G.); davut.dayan@uniklinik-ulm.de (D.D.); inga.bekes@uniklinik-ulm.de (I.B.); Wolfgang.janni@uniklinik-ulm.de (W.J.); rolf.kreienberg@uniklinik-ulm.de (R.K.); 3Universitätsfrauenklinik Würzburg, Josef-Schneider-Str. 4, 97080 Würzburg, Germany; diessner_j@ukw.de (J.D.); Joukhadar_R@UKM.de (R.J.); salmen_j@ukm.de (J.S.); kiesel_m2@ukw.de (M.K.); mueller_m28@ukw.de (M.M.-R.); woeckel_A@ukw.de (A.W.); 4Institut für Medizinische Biometrie, Epidemiologie und Informatik, Universitätsmedizin Mainz, 55131 Mainz, Germany; blettner@uni-mainz.de; 5Helios Amper Klinikum Dachau, Krankenhausstr. 15, 85221 Dachau, Germany

**Keywords:** breast cancer, risk, prediction, BRENDA, score, follow up

## Abstract

**Simple Summary:**

The BRENDA-Score provides an easy to use tool for clinicians to estimate the risk of recurrence in primary breast cancer. The algorithm has been validated via a second independent database and provides five recurrence risk groups. This grouping helps clinicians to encourage high risk patients to undergo the recommended treatment.

**Abstract:**

Background Current research in breast cancer focuses on individualization of local and systemic therapies with adequate escalation or de-escalation strategies. As a result, about two-thirds of breast cancer patients can be cured, but up to one-third eventually develop metastatic disease, which is considered incurable with currently available treatment options. This underscores the importance to develop a metastatic recurrence score to escalate or de-escalate treatment strategies. Patients and methods Data from 10,499 patients were available from 17 clinical cancer registries (BRENDA-project. In total, 8566 were used to develop the BRENDA-Index. This index was calculated from the regression coefficients of a Cox regression model for metastasis-free survival (MFS). Based on this index, patients were categorized into very high, high, intermediate, low, and very low risk groups forming the BRENDA-Score. Bootstrapping was used for internal validation and an independent dataset of 1883 patients for external validation. The predictive accuracy was checked by Harrell’s c-index. In addition, the BRENDA-Score was analyzed as a marker for overall survival (OS) and compared to the Nottingham prognostic score (NPS). Results: Intrinsic subtypes, tumour size, grading, and nodal status were identified as statistically significant prognostic factors in the multivariate analysis. The five prognostic groups of the BRENDA-Score showed highly significant (*p* < 0.001) differences regarding MFS:low risk: hazard ratio (HR) = 2.4, 95%CI (1.7–3.3); intermediate risk: HR = 5.0, 95%CI.(3.6–6.9); high risk: HR = 10.3, 95%CI (7.4–14.3) and very high risk: HR = 18.1, 95%CI (13.2–24.9). The external validation showed congruent results. A multivariate Cox regression model for OS with BRENDA-Score and NPS as covariates showed that of these two scores only the BRENDA-Score is significant (BRENDA-Score *p* < 0.001; NPS *p* = 0.447). Therefore, the BRENDA-Score is also a good prognostic marker for OS. Conclusion: The BRENDA-Score is an internally and externally validated robust predictive tool for metastatic recurrence in breast cancer patients. It is based on routine parameters easily accessible in daily clinical care. In addition, the BRENDA-Score is a good prognostic marker for overall survival. Highlights: The BRENDA-Score is a highly significant predictive tool for metastatic recurrence of breast cancer patients. The BRENDA-Score is stable for at least the first five years after primary diagnosis, i.e., the sensitivities and specificities of this predicting system is rather similar to the NPI with AUCs between 0.76 and 0.81 the BRENDA-Score is a good prognostic marker for overall survival.

## 1. Introduction

As most breast cancer (BC) recurrences occur within the first 60 months [1], after diagnosis, patients have clinical follow-up (*f/u*) visits at 3 months intervals in the first three years, followed by two years with check-ups every 6 months, and then returning to an annual checkup schedule. In BC this is recommended for at least 10 years. This is a well-established schedule in Germany. Several publications have investigated the clinical orientated follow-up versus more intense follow-ups (i.e., including further imaging technology like MRI or tumour markers) [2,3]. In 2016, a Cochrane review on follow-up strategies in early breast cancer did confirm annual mammography and physical exams [4] as sufficient. 

So, thus far, the optimal interval, methods, and parameters have not been determined by prospective randomised trials [2,5]. This might be due to the different health systems, cost and benefit considerations, available resources or the missing survival benefit of an earlier, smaller tumour detection [6]. 

Even though it is known that the pattern of distant metastasis depends on the tumour biology [1,7,8,9,10], primary treatment, and tumour TNM. The follow up recommendations are still ‘one standard fits all’.

To estimate the survival nomograms and prognosis, models have been validated [11,12]. These are used to identify patients with a bad prognosis. If these models are not decisive, gene panels can further differentiate the treatment benefit in defined subgroups. Further treatment recommendations take this into consideration. The same principle is applied for single organ metastasis and local recurrence normograms [13,14], but the clinical consequence is not yet established. 

Following Qi Wu [10], the BRENDA database was used to create a metastatic recurrence index (BRENDA-Index). The individual sum of the BRENDA-Index result in a score (BRENDA-metastatic recurrence score (BRENDA-Score)). This BRENDA-Score identifies the patients’ risk of general metastasis (very high/high/medium/low/very low risk) over time. Ideally, the results would enable the clinician to screen the organ just before or at its highest risk ‘for a recurrence. In order to be a reliable clinical tool the BRENDA-Score should be easy to use with clinical data available at primary diagnosis.

## 2. Materials and Methods

In this retrospective multicenter cohort study, data from the University of Ulm and 16 partner clinics (all certified breast cancer centers) in Baden-Württemberg (Germany) of patients with breast cancer diagnosis between 2000 and 2008 were analysed. Prior analyses of this database named BRENDA (BRENDA breast cancer care under evidence-based guidelines) have been published [7,15,16,17].

This database included a retrospective chart review to extract TNM-stage, histologic subtype, grading, lymphatic and vascular invasion, estrogen/progesterone/erbB-2-expression, date of diagnosis, and all adjuvant therapies. Data on therapies, including surgery (date of surgery, BCT breast-conserving surgery, mastectomy, sentinel-node-biopsy, and axillary lymph node dissection), adjuvant systemic chemotherapy, adjuvant endocrine therapy, and adjuvant radiotherapy, were collected. According to criteria published by Schouten et al. the quality of these data is considered high [18]. Written and informed consent was obtained from all patients included in this study. The inclusion criterion was histologically confirmed invasive breast cancer. The exclusion criteria were carcinoma in situ, primary metastatic disease, bilateral breast cancer, primary occult disease, phyllodes tumour, and patients with incomplete follow-up. As most breast cancer recurrences occur within the first 2 to 5 years patients have follow up visits at 3 months intervals in the first three years, followed by two years with checkups every 6 months and then returning to an annual checkup schedule (optimal intervals, methods, and parameters have not been determined as yet by prospective randomised trials). Although the Kaplan-Meier curves were calculated for a metastatic free survival time of 110 months (approximately 9 years) to show the long term data. The strength of the BRENDA database lies in the detailed information on the actual treatment. Using this a subgroup analysis of the BRENDA-score with guideline-adherent and non-guideline-adherent treatment has been done to show the prediction value regardless of the treatment.

## 3. Surrogate Definition of Intrinsic Subtypes

To define the intrinsic breast cancer subtypes hormone receptor expression (HR), HER2 expression and cell proliferation marker Ki67 were generally used. As Ki67 was not available in the BRENDA database, we used grading as a surrogate parameter to include the cell proliferation, as described before, e.g., by Parise et al. [19], von Minckwitz et al. [20] and Lips et al. [21]. The 5 intrinsic subtypes were defined as follows: Luminal A (HR+/HER2−/grade1 or 2), luminal B-HER2-negative like (HR+/HER2−/grade 3), luminal B-HER2-positive like (HR+/HER2+, all grades); HER2-overexpressing (non-luminal, HR−/HER2+), and triple-negative (basal-like, HR−/HER2−).

## 4. Statistical Analysis

For descriptive purposes, continuous variables were presented as their mean ± standard deviation; skewed variables were presented as their median (interquartile range). Categorical variables were presented as frequencies and percentages. Multiple imputation was used for missing data. Univariate and multivariate analyses using Cox regression were carried out to build a predictive model for metastasis-free survival. All significant factors in the univariate analysis were entered into a multivariate analysis. A prognostic model was established by all factors found to be significantly associated with metastasis-free survival in the multivariate analysis. This model includes intrinsic subtypes, tumour size, grading, and nodal status as “baseline” factors. The beta coefficients of the Cox regression model were multiplied by 10 and rounded to the nearest integer (for 3 parameters with respect to 95% confidence interval of the beta coefficients) to determine the factors multiplying each factor. Taking, e.g., the nodal status N > 3 the beta coefficient is 1.56 and the 95% C.I. 1.35–1.78. Taking into account the 95% C.I. we used 1.5 multiplied by 10. In the case of grading the beta coefficient for G2 was 0.49 (95% C.I. 0.09–1.08) and for G3 0.63 (95% C.I. 0.14–1.23). If we multiply these values by 10 and round to the nearest integer we obtain for G2 5, for G3 6 and the nodal status (N > 3) 16 as weights. Since these values are too high compared to other values, we have reduced them to 4 (G2) and 15 (Nodal status), respectively. These adjustments are each within the 95% confidence intervals. The cutoffs for the grouping were derived by “exhausted chaid” for 5 year metastatic free survival. The proportional hazards (PH) assumption was checked using graphical diagnostics based on the scaled Schoenfeld residuals. For internal model validation we used bootstrap resampling techniques, to provide bias-corrected estimates of model performance, i.e., to obtain stable optimism-corrected estimates. The optimism is the decrease between model performance in the bootstrap sample and in the original sample. The bootstrap results were based on 1000 samples. The item points denote the beta coefficients for covariates in the Cox model, rounded to the nearest integer with respect to 95% CI of betas and multiplied by 10. The metastatic recurrence index (BRENDA-Index) of a patient is the sum of her item points. The BRENDA-Index (range 0–38) was divided into five risk groups (very low ≤ 4; low 5–14; intermediate 15–21; high 22–26 and very high risk ≥ 27) by using exhausted chaid for 5 year metastasis-free survival. These groups define the metastasis recurrence score (BRENDA-Score). Outcome analysis of the BRENDA-Score was performed using Kaplan-Meier estimates and log-rank tests, as well as Cox regression analysis. In order to check whether the prognostic quality is constant over time we used the nearest neighbor estimation (NNE) method for ROC curves from censored survival data [22]. The NNE method guarantees in contrast to the Kaplan-Meier method that sensitivity and specificity were monotone in X for the bivariate distribution function of (X, T), where T represents survival time. In order to test the accuracy, quality and generalizability of this prediction model this model was validated externally with a cohort of 1883 patients (primary diagnosis between 2005 and 2015). The hazard ratios of the derivation and the evaluation sets were compared. In addition, Harrell’s C-index P(Z_i_ > Z_j_|T_j_ > T_i_), for Cox models was calculated (although it is known to be biased) to check calibration, a key component characterising the performance of the prediction model. Calibration is the agreement between prediction from the model and observed outcomes. Furthermore the nearest neighbour estimation was calculated and compared for both Cox models. Because metastasis-free survival (MFS) is a surrogate of overall survival (OS) we compared in addition the BRENDA-Index/Score as a possible prediction tool for overall survival to the Nottingham Prognostic Index/Score [23]. All tests were 2-tailed and statistical significance was defined as *p* < 0.05. Statistical analyses were performed using R (version 3.5) and IBM SPSS Statistics, version 26.0 (IBM Corp., Armonk, NY, USA).

## 5. Results

A total of 8566 patients with primary diagnosis from 2000 onwards were assigned to the development set in this study (Table 1). The median (maximum) observation time was 4.3 y (12.0 y). The median age was 63 years (range, 18–89 years). The median tumour size was 1.9 cm (range, 0.1–28.0 cm). 5.2% (*n* = 443) of the patients had T3/T4 stage tumour. 61.9% (*n* = 5305) were luminal A, 13.8% (*n*=1185) luminal B Her2-negative, 10.2% (*n* = 870) luminal B HER2-positive, 4.8% (*n* = 410) HER2 overexpressing and 9.3% (*n* = 796) triple-negative. Furthermore 38.5% (*n*=3294) were nodal-positive, 28.6% (*n* = 2446) G3 (Table 1) and 1.7% (*n* = 145) M1.

A Cox proportional hazard model was carried out for assessing the association between various clinicopathologic parameters and metastasis-free survival rate. A prognostic model was established by all factors found to be significantly associated with metastasis-free survival in the multivariate analysis. This model includes intrinsic subtypes, tumour size, grading, and nodal status (Table 2). 

Bootstrap validation based on 1000 samples was used to estimate the performance (internal validation) (Table 3).

The beta coefficients of the original model and the bootstrap model were identical. The metastatic recurrence index (BRENDA-Index) of a patient was derived by summing the item points of each prognostic factor in the model. The item points denote the round estimates of beta coefficients for covariates in the Cox model with respect to 95% CI of betas multiplied by 10. 

The BRENDA-Index was calculated using the following formula: BRENDA-Index = 5 ∗ luminal B-HER2-negative like + 4 ∗ luminal B-HER2-positive like +7 ∗ HER2-overexpressing + 8 ∗ triple-negative + 5 ∗ tumour size 2 + 9 ∗ tumour size 3/4 + 4 ∗ grading 2 + 6 ∗ grading 3 + 8 ∗ nodal status (1 ≤ *n* ≤ 3) + 15 ∗ nodal status (*n* > 3). The values of the covariates were 1 if valid, otherwise 0. 

The BRENDA-Index (range 0–38) was divided into five risk groups (very low ≤ 4; low 5–14; intermediate 15–21; high 22–26 and very high risk ≥ 27). These risk groups define the BRENDA-Score. In total, 30.0% of the 8566 patients were very low risk, 31.1% low risk, 20.1% intermediate risk, 9.5% high risk, and 9.4% very high risk. The 5-years metastatic free recurrence rates for the various risk groups were: very low risk 98%, low risk 95%, intermediate risk 90%, high risk 82%, and very high risk 70%. Figure 1 shows the Kaplan-Meier curves stratified by BRENDA-Score. 

Taking the very low risk patients as reference group, we obtained the following hazard ratios (HR) for low risk: HR = 2.4, 95% CI. (1.7–3.3), *p* < 0.001; intermediate risk: HR = 5.0, 95% CI. (3.6–6.9), *p* < 0.001; high risk: HR = 10.3, 95% CI. (7.4–14.3), *p* < 0.001, and very high risk: HR = 18.1, 95% CI. (13.2–24.9); *p* < 0.001.

## 6. External Validation

In total, 1883 patients with primary diagnosis from 2005 to 2015 from the certified breast cancer center in Dachau (Germany) were assigned to the validation set in this study. The median (maximum) observation time was 8.5yr (14.3yr). The median age was 60 years (range, 29–89 yr). In total, 75.7% (*n* = 1425) of the patients were luminal A, 5.9% (*n* = 1112) luminal B Her2-negative, 7.6% (*n* = 143) luminal B HER2-positive, 3.2% (*n* = 60) HER2 overexpressing, and 7.6% (*n* = 143) triple-negative. Furthermore 28.9% (*n* = 545) were nodal-positive, 13.9% (*n* = 261) G3 and all patients M0. In total, 45.6% of the 1883 patients were very low risk with respect to the BRENDA-Score, 30.3% low risk, 14.1% intermediate risk, 5.6% high risk, and 4.4% very high risk. Figure 2 shows the Kaplan-Meier curves of the validation set stratified by BRENDA-Score. 

Taking again the very low risk patients as reference group, we obtained the following hazard ratios (HR) for low risk: HR = 3.1, 95% CI. (2.0–4.6), *p* < 0.001; intermediate risk: HR = 4.5, 95% CI. (2.9–7.1), *p* < 0.001; high risk: HR = 7.7, 95% CI. (4.7–12.7), *p* < 0.001; and very high risk: HR = 14.6, 95% CI. (9.1–23.6); *p* < 0.001.

To validate the predictive ability of the survival model, we calculated Harrell’s c-index (concordance-index), a “global” index for evaluating risk models in survival analysis. Harrell’s c-index (se = standard error): Derivation data 0.77 (se 0.01); derivation data without M1-patients 0.76 (se 0.01) and validation data 0.74 (se 0.02). The Harrell’s c-indexes were all between 0.7 and 0.8 indicating that the BRENDA Index is a good model for metastasis-free survival. 

Next, we calculated time-dependent receiver operating characteristic (ROC) curves at various time points for the derivation and the validation data to evaluate the time-varying performance of the BRENDA-Index by the area under the curve (AUC). We used the nearest neighbour estimation (NNE). The sensitivities and specificities of these predicting systems were rather similar with the area under the curve values falling between 0.76 and 0.81. The AUCs of the BRENDA-Index for 1, 3, and 5 years for the derivation set (evaluation set) were as follows: 1 yr 0.81 (0.76); 3 yr 0.79 (0.78); and 5 yr 0.76 (0.76). This shows that there was close agreement between the AUCs of both datasets for the first five years after primary diagnosis and the values were quite stable.

Metastasis-free survival is a strong surrogate of overall survival. Therefore, the BRENDA-Score should also be a predictor for overall survival. Figure 3 shows the Kaplan-Meier overall survival curves of the deviation set stratified by BRENDA-Score (log rank (Mantel-Cox) test: *p* < 0.001). Taking the very low risk patients as reference group, we obtain the following hazard ratios (HR) for low risk: HR = 1.6, 95% CI. (1.3–2.0), *p* < 0.001; intermediate risk: HR = 2.7, 95% CI. (2.2–3.4), *p* < 0.001; high risk: HR = 4.5, 95% CI. (3.6–5.6), *p* < 0.001; and very high risk: HR = 7.6, 95% CI. (6.2–9.4); *p* < 0.001). There were highly significant differences (*p* < 0.001) between the corresponding overall survival curves (Figure 3).

This result was again externally validated. For the validation set of 1883 patients we obtained the following hazard ratios for overall survival (very low risk is the reference): low risk: HR = 2.0, 95% CI. (1.5–2.8), *p* < 0.001; intermediate risk: HR = 2.5, 95% CI. (1.7–3.6), *p* < 0.001; high risk: HR = 4.3, 95% CI. (2.8–6.4), *p* < 0.001; and very high risk: HR = 8.2, 95% CI. (5.6–12.1); *p* < 0.001. Harrel’s c-index for overall survival is 0.688 for the derivation data and 0.715 for the evaluation data [24].

Finally, the BRENDA-Score was compared to the classic Nottingham prognostic score (NPS), a very good prognostic marker for overall survival [12]. A multivariate Cox regression model for overall survival with BRENDA- and Nottingham prognostic score (NPS) showed that in this Cox model only the BRENDA-Score is significant (BRENDA-Score *p* < 0.001; NPS *p* = 0.447). Therefore, the BRENDA-Score is at the moment a good prognostic marker for overall survival.

## 7. Discussion

Once the initial short term breast cancer treatments with surgery, radiotherapy, and systemic treatment are finished, most patients receive a course of anti-hormonal or antibody treatments for a longer time. Despite these treatments this is the transition period into the follow up time. The national guidelines initially recommend a clinical exam every three months for two to three years, followed by bi-annual exams and visits, and then after 5 years annual controls. Mammography and breast ultrasound should be used, alternating yearly [25].

Unfortunately there were no prospective randomized trials, nor trials showing a benefit for more intensive follow up examinations [25] in the general breast cancer population. This may be due to the heterogeneous disease which breast cancer is and the consecutive varying metastasis pattern. Various publications have published risk factors and survival analyses according to the known variables [26,27,28].

Using administrative data to estimate the cancer recurrence was systematically reviewed by Izci et al. [29]. They report of 17 articles with accuracy averaging at 92.2% (95% CI 88.4% to 94.8%). The results show the need for more standardisation and validation of such models. A combination of rule-based approaches with machine learning algorithms is seen as an interesting approach. 

A review [30] found 58 models which predicted mortality and recurrence for breast cancer patients. Only 17 models were externally validated by comparing the predicted outcomes with observed outcomes. Some of these models have very good predictive values in a well-defined subgroup. For example Huang et al for a Taiwan population [31]. Though validated with a large external database, prognosis for Black and Caucasian was significantly under- or overestimated at certain follow up periods.

Another model called PREDICT is available in its original form and an updated version [32]. The update improved the figures for young breast cancer patients significantly but an external evaluation with three databases left the overall AUC at 0.752 for the update. This was—overall—not significantly better than the AUC of the original version.

Widely known and used is the Nottingham prognostic index (NPI) by Haybittle [33]. Initially designed for primary operable breast cancer it takes tumour size in mm, nodal stage and tumour grading into considerations. Over time it has been improved by adding further prognostic factors and differentiating into more subgroups [12]. Due to its wide usage and evaluation in various populations it may be considered the gold standard of prognosis estimation. 

In 2010, Sanghani et al. [34] published the IBTR 1.0 for 10 year ipsilateral breast recurrences. Though this is well validated and improved to version 2.0 by Kindts [35] in 2016 the clinical relevant recurrences were more likely the distant metastasis as local recurrence might be palpated or found via routine mammography or ultrasound. Witteveen published a similar model in 2015 [36] for local recurrences. Here an estimation is provided for year 1, 3, and 5 after initial treatment. Witteveen herself proposed an age and risk adapted follow up to improve the outcome for patients and health care providers. Her data suggests benefits by additional visits around the second year but otherwise twice annual examinations. The focus of this study was again on local recurrences and does not take other distant metastasis into account. Based on this model for local recurrence Draeger et al simulated an individualised follow up [37] on a historic population. This resulted in less patient visits, cost savings, and a delay in diagnosis of local recurrence, but earlier detection of local recurrence might benefit the survival of the patients [38]. So the question regarding the overall benefit remains unanswered.

Our model is based on a European Caucasian population and real world data. It is known that survival depends on guideline adherent treatment and, therefore, this may influence the model calculation. The BRENDA database consists of high quality data from certified breast centers and guideline adherence is a quality indicator monitored regularly in those centers. Further, the BRENDA database was designed to evaluate the guideline adherence. This could result in an ‘ideal’ world model regarding treatment status of the survival data. Of course, such a database cannot contain every current variable with a long term follow up. Parameters like complete pathological response, tumour stage, post-neoadjuvant systemic treatment or the latest immune therapy options can be used to improve the BRENDA-Score once sufficient numbers and sufficient follow-up data are available. As with each grouping of variables, information is lost the tumour stage is used via the original parameters at that time (tumour seize, nodal status, and distant metastasis). 

Special subgroups, like TNBC or young patients, have not been explicitly tested. Mainly because the BRENDA database consists of real world data with an epidemiological mixture of breast cancer patients. Any subgroup testing from within this database can statistically not contribute to the improvement of the model. As there were various subgroups possible the authors encourage scientists to validate the model with their specific patient subgroup. 

However, some of those weaknesses might be just theoretical and may not be clinically relevant. The model was created using high quality data. The database contains unused information on guideline adherent treatment and high quality follow-up. The effect of guideline-adherent treatment on survival in this dataset has been published prior [16]. This information was not taken into consideration during the primary analysis for the development of the index As the BRENDA-score is a clinical tool used at the time of diagnosis to show the long term risk of recurrences this risk can be lowered by guideline-adherent treatment [16]. To strengthen the predictive value of the model it was evaluated via two different methods. First, a bootstrap evaluation was done and confirmed the initial results. The second evaluation with an external independent database of a certified breast center reproduced again highly significant outcomes and the initial prognostic value of our model. Time dependent ROCs were calculated and compared to published ROC results by Sejben et al. [39]. The authors determined for TNBC the ROC for the NPI, PrognosTILs and Predict model. The NPI was the highest with 0.781. Here our results were very similar.

As mentioned before, a nomogram can enable clinicians to identify the high risk patients and ensure the optimal treatment. In order to individualise the *f/u* the model needs to be more precise. Ideally, the results would indicate the time period for the organs at the highest risk, enabling the clinician to monitor this organ more intensively (i.e., blood sample or imaging). However, thus far, a more detailed model is needed. Additionally, this theoretic individualisation of the *f/u* needs to be evaluated for its patients and health care cost benefit in clinical trials. The current S3 guideline points out the lack of evidence regarding individualised follow up [25]. 

Our model seems to be superior to the NPI for risk of recurrence in distant organs. Even though new methods, like liquid biopsy, were published [40,41] clinically those methods were not available widely. Despite the unavailability for most caretakers the cost and benefit is still unknown. Here the BRENDA-Score might be helpful. Sparano et al. [42] proposed the implementation of ‘liquid biopsy’ or circulating tumor cells in prospective follow-up. Considering the cost and unknown benefit of such tools our algorithm could be used to identify patients with a very high recurrence risk for study participation, thus helping the conducting scientists to optimise the available resources and reducing the number patients needed to recruit and possibly the follow up time. With a risk distribution per primary tumour data the clinician could easily identify patients at risk and could monitor the patient more closely or encourage the participation in studies evaluating surveillance methods. The authors encourage the validation of the BRENDA-Score on different subgroups and research questions.

## 8. Conclusions

In this retrospective study, with a derivation set of 8566 patients and an evaluation set of 1883 patients, all from certified breast cancer centers, a metastatic recurrence score (BRENDA-Score) and respective index (BRENDA-Index) were developed and internally and externally evaluated. The main findings of this study were:(1).The BRENDA-Score is a highly significant predictive tool for metastatic recurrence of breast cancer patients;(2).It is based on routine parameters, easily accessible in daily clinical care;(3).The BRENDA-Score is stable over at least the first five years after primary diagnosis, i.e., the sensitivities and specificities of this predicting system is rather similar with AUCs between 0.76 and 0.81;(4).Internal and external validations confirmed these results;(5).Finally, the BRENDA-Score is in addition a good prognostic marker for overall survival. This confirms that metastatic free survival is a strong surrogate parameter for overall survival;(6).A multivariate Cox regression model for overall survival with BRENDA- and Nottingham prognostic score (NPS) showed that only the BRENDA-Score is statistically significant.

## Figures and Tables

**Figure 1 cancers-13-03121-f001:**
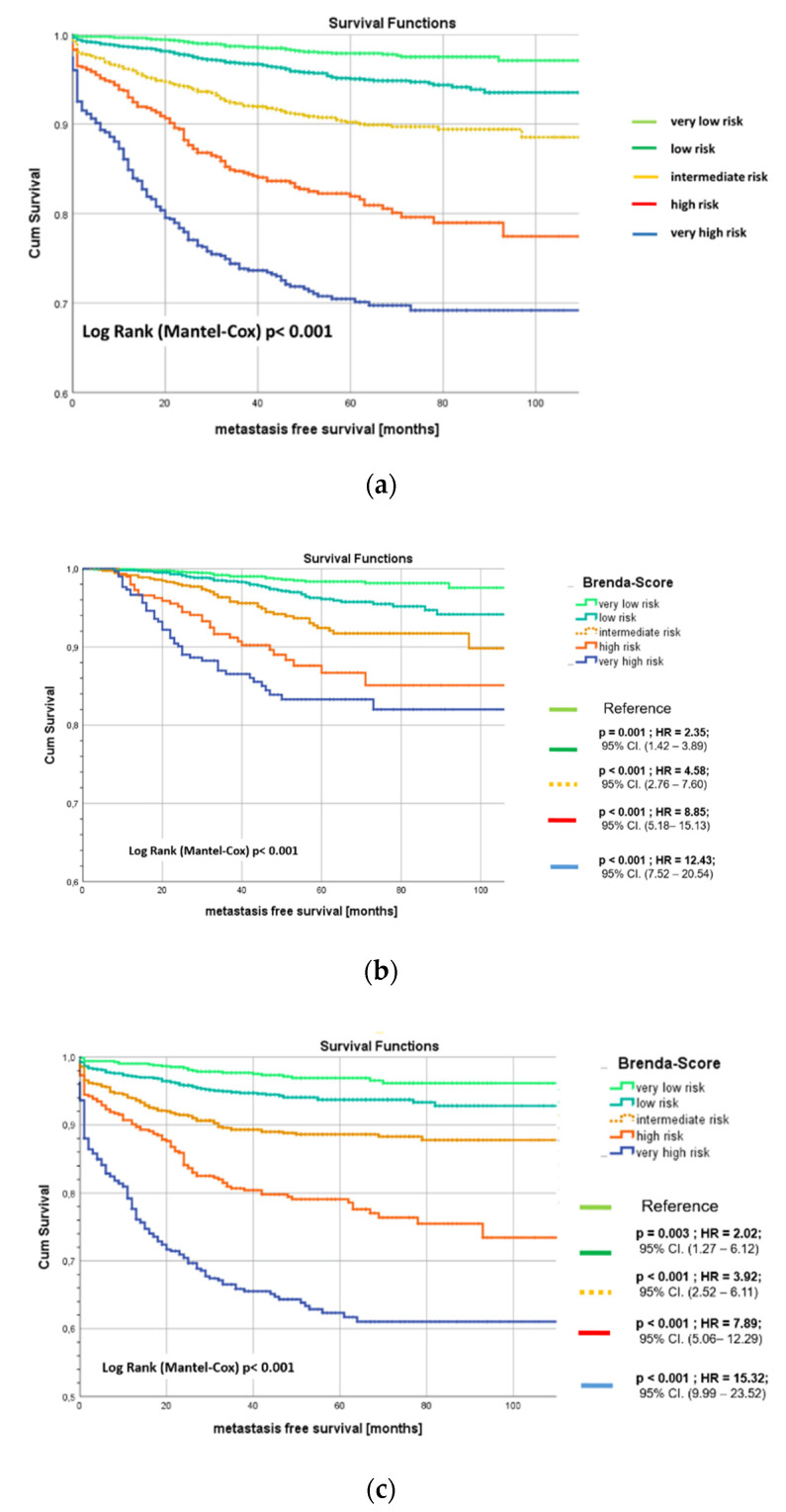
(**a**) Kaplan-Meier metastasis-free survival (MFS) curves of the derivation set stratified by BRENDA-Score (BRENDA DB) with 110 months. Very low risk = light green, low risk = green, intermediate risk = yellow, high risk = orange, very high risk = blue line. (**b**,**c**) Subgroup analysis with guideline adherent (**b**) and non-adherent (**c**) treatment Kaplan-Meier metastasis-free survival (MFS) curves of the derivation set stratified by BRENDA-Score (BRENDA DB) with 110 months. Very low risk = light green, low risk = green, intermediate risk = yellow, high risk = orange, very high risk = blue line.

**Figure 2 cancers-13-03121-f002:**
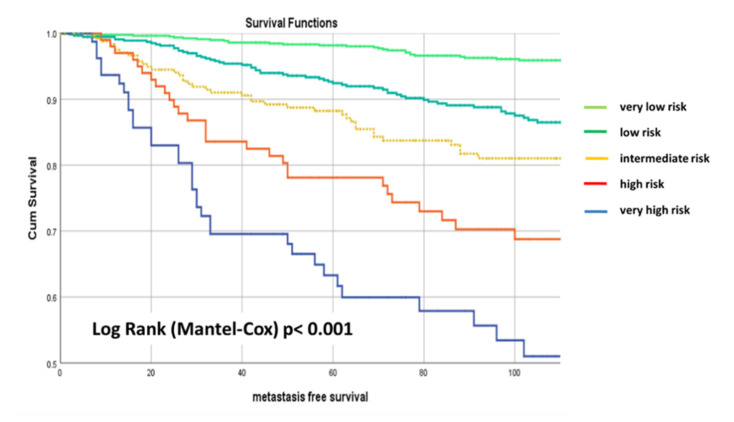
Kaplan-Meier metastasis-free survival (MFS) curves of the validation set stratified by BRENDA-Score (Dachau DB) with 110 months follow up. Very low risk = light green, low risk = green, intermediate risk = yellow, high risk = orange, very high risk = blue line.

**Figure 3 cancers-13-03121-f003:**
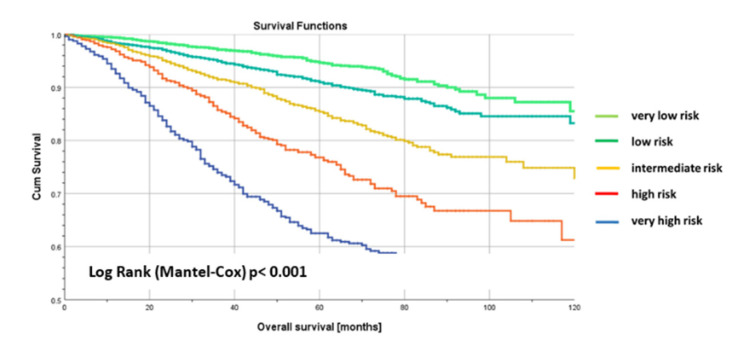
Kaplan-Meier curves of overall survival curves of the deviation set stratified by BRENDA-Score (BRENDA DB) with 10 years follow up. Very low risk = light green, low risk = green, intermediate risk = yellow, high risk = orange, very high risk = blue line.

**Table 1 cancers-13-03121-t001:** Basic characteristics.

	Brenda-Score						
	Total	Very Low Risk	Low Risk	Intermediate Risk	High Risk	Very High Risk	
	*n* = 8566 (%)	*n* = 2568 (30.0)	*n* = 2661 (31.1)	*n* = 1718 (20.1)	*n* = 817 (9.5)	*n* = 802 (9.4)	Sig
	62.1 ± 13.1 min 18; max 89	62.1 ± 13.1 min 18; max 89	61.5 ± 13.2 min 24; max 89	62.6 ± 14.0 min 27; max 89	62.9 ± 13.7 min 18; max 89		0.005
T1	4842 (56.5)	2568 (100)	1616 (60.7)	428 (24.9)	178 (21.8)	52 (6.5)	<0.001
T2	3281 (38.3)	0 (0)	988 (37.1)	1192 (69.4)	606 (74.2)	495 (61.7)	
T3/T4	443 (5.2)	0 (0)	57 (2.1)	98 (5.7)	33 (4.0)	443 (5.2)	
nodal negative	5272 (61.5)	2569 (100)	2063 (77.5)	626 (36.4)	15 (1.8)	0 (0)	<0.001
1–3 affected lymph nodes	1974 (23.0)	0 (0)	598 (22.5)	903 (52.6)	362 (44.3)	111 (13.8)	
>3 affected lymph nodes	1320 (15.4)	0 (0)	0 (0)	189 (11.0)	440 (53.9)	691 (86.2)	
G1	817 (9.5)	539 (21.0)	228 (8.6)	7 (0.9)	7 (0.9)	1 (0.1)	<0.001
G2	5303 (61.9)	2029 (79.0)	1755 (66.0)	367 (44.9)	367 (44.9)	187 (23.3)	
G3	2446 (28.6)	0 (0)	678 (25.5)	443 (54.2)	443 (54.2)	614 (76.6)	
luminal A	5305 (61.9)	2552 (99.4)	1559 (58.6)	790 (46.0)	316 (38.7)	88 (11.0)	<0.001
luminal B-HER2-negative like	1185 (13.8)	0 (0)	315 (11.8)	353 (20.5)	268 (32.8)	249 (31.0)	
Luminal B-HER2-positive like	870 (10.2)	16 (0.6)	377 (14.2)	236 (13.7)	95 (11.6)	146 (18.2)	
HER2 overexpressing	410 (4.8)	0 (0)	126 (4.7)	109 (6.3)	60 (7.3)	115 (14.3)	
Triple-negative	796 (9.3)	0 (0)	284 (10.7)	230 (13.4)	78 (9.5)	204 (25.4)	

**Table 2 cancers-13-03121-t002:** Cox regression model for metastasis-free survival including intrinsic subtypes, tumour size, grading and nodal status. B beta coefficient; SE standard error; Sig. significance; HR hazard ratio; CI confidence interval with lower and upper limit.

Variables in the Equation						
Covariates	B	SE	Sig.	HR	95% Confidence Interval	
					lower	upper
luminal B-HER2-negative like	0.48	0.18	0.007	1.61	1.14	2.28
Luminal B-HER2-positive like	0.42	0.15	0.004	1.53	1.14	2.04
HER2 overexpressing	0.72	0.18	0.000	2.06	1.45	2.94
Triple-negative	0.76	0.17	0.000	2.14	1.55	2.96
T2	0.50	0.09	0.000	1.65	1.38	1.97
T3/T4	0.96	0.14	0.000	2.62	2.01	3.41
G2	0.49	0.24	0.038	1.64	1.03	2.61
G3	0.63	0.27	0.020	1.89	1.10	3.22
1 ≤ *n* ≤3	0.80	0.11	0.000	2.23	1.81	2.74
*n* > 3	1.56	0.10	0.000	4.75	3.89	5.79

**Table 3 cancers-13-03121-t003:** Bootstrap results based on 1000 bootstrap samples (internal validation) B beta coefficients.

Bootstrap for Variables in the Equation						
	B	Bootstrap ^a^				
		Bias	Std. Error	Sig. (2-Tailed)	95% Confidence Interval	
					lower	upper
luminal B-HER2-negative like	0.48	0.01	0.18	0.010	0.11	0.83
Luminal B-HER2-positive like	0.42	−0.01	0.15	0.005	0.09	0.70
HER2 overexpressing	0.72	−0.01	0.18	0.001	0.36	1.05
Triple-negative	0.76	0.00	0.17	0.001	0.42	1.07
T2	0.50	0.00	0.09	0.001	0.32	0.69
T3/T4	0.96	−0.01	0.14	0.001	0.66	1.22
G2	0.49	0.03	0.25	0.034	0.09	1.08
G3	0.63	0.02	0.28	0.014	0.14	1.23
1≤ *n*≤ 3	0.80	0.00	0.10	0.001	0.60	1.00
*n* > 3	1.56	0.00	0.11	0.001	1.35	1.78


^a^ Unless otherwise noted, bootstrap results are based on 1000 bootstrap samples.

## Data Availability

Data is available upon reasonable request via the BRENDA study group.

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
