# Peer review of "BRENDA-Score, a Highly Significant, Internally and Externally Validated Prognostic Marker for Metastatic Recurrence: Analysis of 10,449 Primary Breast Cancer Patients"

_cancers, 2021, doi:10.3390/cancers13133121_

Round 1

Reviewer 1 Report

In this manuscript, the authors used a BRENDA-score to predict patient outcome. Using multivariate cox model, they built a risk score model and classified patients into different groups. KM curves showed significant difference in patient survival as expected. The results are quite interesting. However, several issues as follows the authors might have addressed could improve the quality of the presentation.

  1. In calculating the risk score, the authors used the beta of variables in cox model. Given rounding up the multiplication of beta and 10, why some variables, e.g., nodal status 15, and tumor size 3/4 9 not 16 and 10? In addition, it is unclear how the authors determine the cutoffs for the grouping.
  2. The authors proposed three criteria for BRENDA score, however, the first and third one have not been well presented. They mentioned the sensitivities and specificities, a table is expected to present   such data in the results. For the third one, the AUC, sensitivities and specificities of two methods are expected to be compared with each other.  
  3. Why did the authors not include disease stage information in the model?
  4. How many patients received chemo-, endocrine-therapy? Will the results be held if considering the treatment? 
  5. The model is quite stable for 5 yrs as the authors concluded. Will it  be still valid for 10 yrs survival or longer?

Reviewer 2 Report

The manuscript presents results of a new scoring system (BRENDA-score) to calculate the risk of recurrence and distant metastases in primary breast cancer patients. The Brenda score is based on standard pathological parameters such as the TNM stage, receptor- and HER2-status and primary therapy. Dividing patients into five groups (very low, low, intermediate, high, and very high risk), this scoring system is reported to be highly prognostic for distant recurrence and overall survival in retrospective analysis, and even superior to the Nottingham Prognostic Score (NPS).

Up to now, standard survivorship care does not take into account the individuals` risks, and recommendations are the same for all patients. The search for methods to individualize follow-up strategies, and intensify follow-up examinations in case of higher risk, has started back in the 1980ies. However, even intensified follow-up strategies based on risk estimations could not demonstrate therapeutic benefit regarding overall survival of patients. Nevertheless, new treatment options have arisen in the past years to improve secondary adjuvant therapy or treatment of metastatic disease. Therefore, results of this study might be relevant to improve prognostic estimation and individualize the surveillance of patients.

Some minor revisions should be made:

  1. Language and formate: The manuscript should be proofread for English language and for formate
  2. Discussion: Newer methods for prognostic estimation beneath standard pathologic parameters, such as gene panels (Oncotype, Endopredict etc.) should be mentioned. Furthermore, future methods of surveillance, like liquid biopsy (CTC, cfDNA etc.) could potentially be integrated into follow-up. There is a strong need for prospective follow-up trials taking into acount this newer methods, as proposed by Sparano et al. (Sparano, JNCI 2019). This should be discussed.

Round 2

Reviewer 1 Report

Thank the authors for revision. I do not get the answer on the question 1. I can understand 0.49 x10=4.9, round to 5, and 0.63x10=6.3, round to 6, however, I do not understand 1.56x10=15.6,  round to 15, why use 1.5x10, not 1.56x10? Second, I do not understand the cutoff of  "exhausted chaid" as you mentioned. If based on the reference(s), please provide. Otherwise, a detail information is preferred. The answers to other concerns are ok. 

Author Response

Dear reviewer,

thank you for your question on more clarification. A scoring system like our BRENDA-score derived on an additive scale (i.e. regression coefficient) requires additive weights. There are several scoring algorithms to derive the corresponding weights (Beta, Beta/Integer; Beta10/integer; Beta/Schneeweiss, Beta /Sullivan,…). We used Beta10/integer. Additionally one can affiner the above weights within the 95% CI of the Beta coefficients taking into account some clinical background knowledge.

Taking into account the clinical background of the risk of our significant parameters, 2 weights were too high in relation to the weights of other risk factors. As we mentioned before, Beta10 / integer returns the value 5 for G2 and the value 6 for G3. It is known from numerous publications that this distance must be greater. We have therefore corrected the value for G2 from 5 to 4. The same applies to the nodal status (N> 3). Beta10 / integer returns the value 16, a value that is too high in relation to other weights. We have therefore (very moderately) reduced it to 15.   We have now inserted a sentence on this into the publication: „The calculated weights for G2 and the nodal status (N> 3) are 5 and 16. Since these values are too high compared to other values, we have reduced them to 4 and 15, respectively. These adjustments are each within the 95% confidence intervals“   We used exhaustive CHAID to calculate cut-off values for the BRENDA index. We took 5 years of metastasis-free survival as the target value, since metastases are known to occur essentially within the first 5 years (as a result of the tumor doubling time of metastases).   Exhaustive CHAID, a modification of the basic CHAID algorithm, performs a more thorough merging and testing of predictor variables.  More information can be found e.g. in Tzung-I Tang et al.: A Comparative Study of Medical Data Classification Methods Based on Decision Tree and System Reconstruction Analysis , IEMS Vol. 4, No. 1, pp. 102-108, June 2005. Yours, Florian Ebner